# E-Commerce Parcel Distribution in Urban Areas with Sustainable Performance Indicators

**Riharsono Prastyantoro [1,\*], Heru Purboyo Hidayat Putro [2], Gatot Yudoko [2] and Puspita Dirgahayani [2]**

[1] Graduate Program in Transportation, School of Architecture, Planning and Policy Development, Bandung Institute of Technology, Bandung 40132, Indonesia
[2] School of Architecture, Planning and Policy Development, Bandung Institute of Technology, Bandung 40132, Indonesia
\* Correspondence: soni.riharsono@gmail.com

**Abstract:** E-commerce transactions have increased during the pandemic as people living in urban areas turn to buy goods online rather than offline. A two-echelon distribution system using parcel mobile hubs (PMHs) with small vehicles can increase operational cost efficiency and reduce greenhouse gas emissions. The originality of this study is in building and testing a conceptual framework for selecting PMH locations with three variables (parcel distribution, internet quota for e-commerce, and center of e-buyer) and two constraints (space availability and traffic flow). Spatial analysis is used as a method to test the conceptual framework with a parcel distribution database from Bandung. As a city whose profile represents urban areas in developing countries well, Bandung is chosen as a case study. The proportion of distance to variables and the outermost point of each cluster is less than 10%, which proves that the three variables are correlated. This study proves that the selection of PMH locations based on this conceptual framework results in better sustainable performance compared to existing conditions. Using PMHs combined with city freighters can reduce operational costs by 19.7% and prevent 3.4 tons of $CO_2$ emissions per year with conventional motorcycles and 7.2 tons of $CO_2$ emissions per year with electric motorcycles or scooters.

**Keywords:** mobile hub; urban freight transportation; e-commerce parcel; spatial analysis; sustainable performance index

## 1. Introduction

The gross transaction value of e-commerce in Indonesia was USD 20.9 billion in 2021 and is expected to reach USD 82 billion by 2025, which is an enormous value in Southeast Asia [1]. The COVID-19 pandemic has changed consumer lifestyles and encouraged the use of e-commerce [2]. As the number of e-commerce transactions increases, the challenges associated with parcel distribution also increase, where congestion and accessibility are crucial factors [3].

The most significant GHG (greenhouse gas) emissions come from the transportation sector, which contributes more than 18.7 million tons of $CO_2$ per year. City governments are interested in ensuring the smooth distribution of goods while maintaining or reducing $CO_2$ emission levels [4]. The iPrice research team conducted a study with 80,000 respondents from five ASEAN countries. One of their findings was that more than 90% of customers submitted complaints and negative responses regarding delivery delays [5]. Courier, express, and parcel (CEP) operators must find alternative solutions at every stage, including demand forecasting, inventory management in warehouses, technology integration, distribution systems, and fast delivery to end consumers [6].

One of the most expensive, least efficient, and most polluting parts of the entire logistics chain is last-mile delivery carried out in urban areas [7], which accounts for approximately 28% of total transportation costs [8]. Innovation in last-mile delivery will significantly contribute to improving efficiency and reducing GHG effects.

This study was motivated by the need to determine the organizational form of the ideal distribution system for delivering e-commerce parcels and the location of PMHs, as well as to develop a conceptual framework for selecting the best site for PMH distribution in urban areas in terms of green performance.

### 1.1. Characteristics of E-Commerce Parcels

Most online parcels are delivered to consumers in urban areas [3]. In Indonesia, these consumers are concentrated in urban areas on Java island, which accounts for 75.77% of all e-commerce consumers [9]. The increase in e-commerce transactions directly impacts the urban logistics system, where the level of congestion and accessibility are crucial factors [3]. The city government is interested in ensuring the smooth distribution of goods while maintaining or reducing levels of $CO_2$ emissions and other pollutants [4]. City councils in other countries require the use of environmentally friendly light vehicles, such as cargo tricycles or electric bicycles, to pick up and deliver parcels in Atlanta [10], Rio De Janeiro [11], and Brussels [12]. During the COVID-19 pandemic in India, trucks combined with drones were used for social assistance delivery to beneficiary communities [13].

E-commerce parcels have unique characteristics compared to regular parcels (Table 1). E-buyers demand faster and cheaper delivery services.

**Table 1.** Characteristics of regular parcels and e-commerce parcels.

| Characteristics | Regular Parcels | E-Commerce Parcels |
| --- | --- | --- |
| Business Growth | 10–12% in 2021 [14] | 33% with a market value of IDR 337 trillion in 2021 [15] |
| Subject | Parcel sender | Customer/e-buyer/parcel receiver |
| Price and weight | variation $\leq$ 30 kg | Item price between IDR 100 k–500 k with weight around 2.2 kg/small parcel [16] |
| Delivery Destination | Nationwide | 75.77% in urban Java [9] |
| Delivery Destination Concentration | Scattered | There are e-buyer areas such as student dormitories, settlements, and offices [17] |
| Delivery Time | Standard, express, premium | Express, premium/demanding [18] |
| Service Availability | Workdays, office hours | Non-stop or 24/7 [18] |
| Price | Market Mechanism | Market Mechanism + startup intervention [19] |

### 1.2. Two-Echelon Distribution System with Mobile Hub

The last mile, the final leg of a distribution sequence that commonly links a distribution center and a customer (store), is a challenge for most freight distribution networks, particularly in retail [20]. Last-mile delivery (LMD) is the last journey where the product leaves the distribution center before it reaches the customer. This strategy can increase the efficiency of distribution operations because the vehicle size can be adjusted according to the number of parcels sent [21]. According to [8], last-mile delivery accounts for more than 28% of the total transportation cost. The strategy of distributing various kinds of parcels to customers through intermediaries is an increasingly popular strategy in logistics to reduce the environmental (energy use and congestion) and social (air pollution, accidents, and sources) consequences of logistics operations [22]. CEP operators often use a two-level distribution system [23].

Meanwhile, [24] uses the term two-echelon distribution system to describe a system in which parcels are consolidated at the distribution center (first echelon) and sent to a satellite/hub/depo (second echelon), after which they are then delivered to the customer's address. Urban logistics aims to optimize sophisticated urban transportation systems through urban logistics planning with integrated short-term operation scheduling and resource management involving a two-echelon distribution system structure [25]. The most critical infrastructure for two-echelon distribution is the satellite, which is what allows

for the transshipment and consolidation of parcels that are delivered to customers [26]. The resources in the two-echelon distribution system are urban truck satellites and city freighters. Determining the most efficient satellite location is the key to consolidation and coordination [25].

Ducret [27] divides logistics organizations into classic and innovative ones for express parcel delivery. CEP operators must adopt new services and innovative delivery tools for customers' benefit [28]. Mobile hubs, as part of an innovative logistics organization, are widely used by CEP operators through innovation and operational efficiency to offer services at competitive rates [27]. Last-mile delivery that combines mobile hubs and small vehicles has become an innovative solution in urban areas to realize a sustainable transportation system [11,12,29]. Anagnostopoulos and Zaslavsky [30] proposed a mobile concept in the form of large trucks integrated with small trucks to reduce congestion in smart cities. Several articles have examined distribution systems with mobile hubs that combine small vehicles to solve urban challenges. The small vehicles are electric cargo bikes [10,12,29], cargo tricycles [11], drones [31–33], and low-capacity trucks (LCTs) integrated with the IoT [30].

Appendix A provides a research overview on modeling a 2E-VRP distribution system using a mobile access hub (MAH). Some researchers have proposed the use of city freighters in the form of electric cargo tricycles [10–12,29,34] and drones [31,32,35]. In developing countries, where the electric vehicle ecosystem has not been formed properly, it is necessary to supplement city freight with non-electric small vehicles. We developed a distribution system for city freight that involves fossil fuel motorbikes, but in the future, they can be replaced by electric motorcycles or scooters.

Researchers have previously proposed distribution systems for various commodities, including garbage [33], social subsidies or services [35,36], and parcels [10,12,29]. We developed a distribution system for e-commerce parcels that have different characteristics from regular parcels (Table 1).

Some researchers have measured their distribution systems using sustainable indicators [10,12]. Verlende [12] measured the social aspects via interviews with twelve random people who represented the community. Meanwhile, [10] measured the social aspects in terms of minimizing the wait time. We propose to measure the social aspects via interviews with thirty deliverymen.

Various e-commerce challenges that occur, as well as previous research related to two-echelon distribution systems, raise the question of how to build a distribution system for e-commerce parcels in urban areas with sustainable performance indicators.

The originality of this study is the development and testing of a conceptual framework for selecting PMH locations for e-commerce parcel distribution. The proposed new variables are the use of internet quota for e-commerce and centers of e-buyers in the form of offices, student dormitories and apartments. The test was carried out in two stages, namely, testing the correlation between the conceptual framework variables. The second test is to compare the measurement of transportation performance indicators between the innovative distribution system and PMH with the classic distribution system.

## 2. Materials and Methods

This study aims to develop a distribution system for e-commerce packages in urban areas with sustainable indicators. The objective of this research is to determine the distribution system for e-commerce parcels, develop a conceptual framework for selecting MAH locations, and calculate sustainable transportation indicators.

The method used in this study is a mixed method combining quantitative and qualitative methods according to the characteristics of the research stages to fully understand the studied reality [37]. The quantitative method is in the form of spatial analysis, optimization, and heuristics, while the qualitative method is in the form of in-depth interviews with a sample of deliverymen. The stages of the research are described as follows:

a.  Determination of the distribution system. E-commerce parcels have specific characteristics, in contrast to regular parcels (Table 1) or other commodities, so it is necessary to determine the most effective and efficient distribution system;

b.  Selection of PMH location. We build a conceptual framework for determining the best location for PMHs to distribute parcels. The method used is a spatial analysis by calculating the center of gravity of the research variables;

c.  Determination of PMH vehicle. The PMH vehicles used need to be determined in terms of type, capacity, and number in accordance with the volume of packages;

d.  Routing and scheduling. We use the Clark and Wright algorithm (saving algorithm) to create a cluster and then use the optimization model to determine the optimal route;

e.  Calculation of sustainable performance indicators. The routing and scheduling results determine the distance of PMHs from the destinations and thus the city freighter travel distance, which is then used to calculate the transportation performance in terms of its economic, social, and environmental aspects. The results of routing and scheduling calculations are then used to calculate economic and environmental indicators. The social indicators were measured through sampling interviews with 30 courier operators.

### 2.1. Establishment of a 2-Echelon Distribution System Organization for E-Commerce Parcels

Based on the results of previous research related to a 2-echelon distribution system with MAH (Appendix A), we propose a 2-echelon satellite-synchronized capacitated vehicle routing problem parcel mobile hub simultaneous pickup delivery (2E-SS-CVRP-PMH-SPD) system, which is illustrated in Figure 1. This 2-echelon distribution model uses a city hub as a parcel mobile hub (PMH) combined with a city freighter in the form of a small vehicle with simultaneous pickup and delivery.

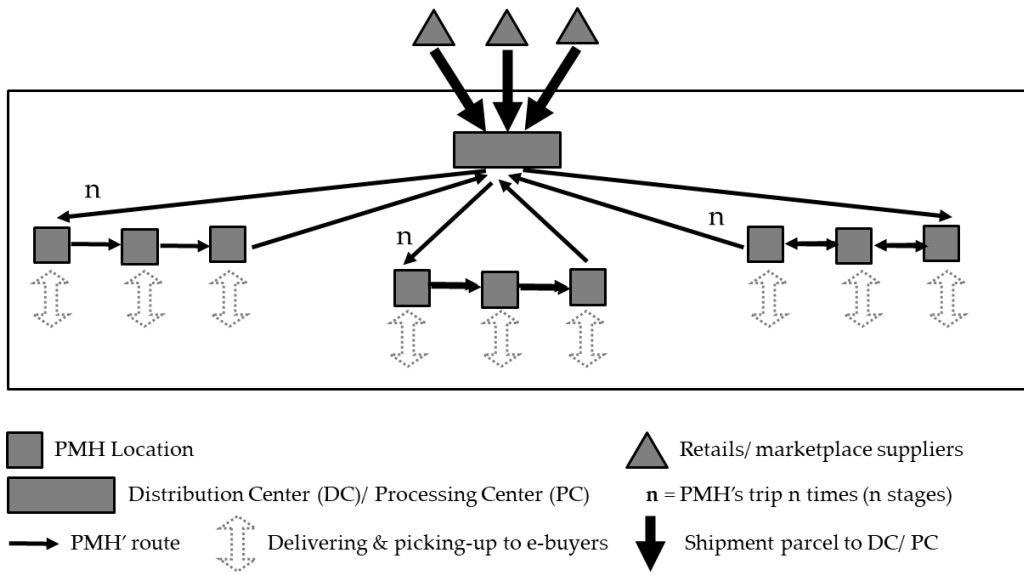

**Figure 1.** Innovative logistic organization with PMH: 2E-SS-CVRP-PMH-SPD. Adapted from [27].

The distribution process of the two echelons with PMHs in each cluster operates according to the following sequence:

1.  The PMH departs from the distribution center to the PMH location (phase I) according to the predetermined route and schedule;

2.  The PMH arrives at the location to hand over the parcel (unloading) to the deliverymen according to their respective delivery zones;

3.  The officer delivers the parcel to the customer's address within the designated delivery area within a maximum of 4 h;

4.   The PMH continues the journey to the next location (point b), then proceeds until the last location before returning to the distribution center;
5.   The PMH departs from the distribution center to the PMH location (phase II) to pick up parcels and hand them over to the deliveryman for the second delivery;
6.   This process is repeated for n stages as planned.

### 2.2. Construction of a Conceptual Framework for the Location Choice for PMHs

Determining the most efficient satellite location is the key to consolidation and coordination [25]. Regarding e-commerce parcel delivery schemes, little is known about the receiving point of proximity and location criteria, travel chain patterns, and the tracking and tracing of information communication technology (ICT) tools [3]. Verlinde et al. [12] placed a mobile hub in the middle of the city of Brussels, and emissions of pollutants dropped significantly (a 24% decrease in $CO_2$ emissions and a 99% decrease in PM2.5 emissions). In contrast, other researchers have determined the location of a mobile hub based on the centroid of a zone or the midpoint between 2 zones [10].

The dynamics of the facility location problem (FLP) must be considered when modifying the current facility or developing a new facility to efficiently handle changing parameters such as market demand, external and internal factors, and population [38]. Mobile facilities can be placed in different locations to offer services to nearby events.

Some researchers determine the location of a MAH using the center of the zone without regard for the distribution of the parcels delivered in Brussels [12] and Atlanta [10]. We determine the locations of PMH that considers the distribution of the parcels delivered.

We build a conceptual framework for selecting PMH locations based on the synthesis of previous studies involving centroid zones in Brussels [10], location determination for pickup point networks (PPNs) in Seine-et-Marne [3], and e-buyer areas in urban areas based on land use categories such as student dormitories in Southampton [17], residences, and offices in Berlin [39], and the results are shown in Figure 2.

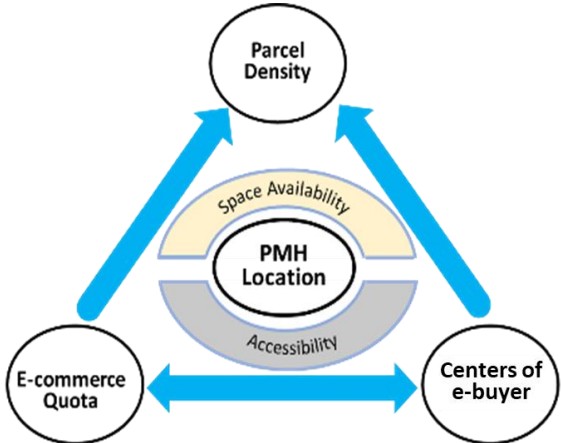

**Figure 2.** Selection of PMH locations for e-commerce parcel delivery in urban areas.

The selected PMH location is the best location with the most efficient/cheapest transportation costs. The hypothesis is that the location of PMHs is influenced by 3 spatial variables, namely parcel distribution (parcel density), the demographics of quota usage for online shopping (e-commerce quota), and e-buyer areas in apartments, offices, and student dormitories (centers for e-buyers). The parcel demand rate in a region is influenced by the quota usage rate and the density of e-buyers in offices, apartments, and student dormitories.

Parcel density is a historical variable, while the use of quotas and e-buyer areas are considered potential factors affecting parcel density in the future, which is why the relationships are depicted by a 1-way arrow. In contrast, the 2-way arrows show that the variables of quota usage and e-buyer areas influence each other. The PMH location is also influenced by two main constraints: the availability of space and the level of traffic flow.

The courier operator can use the selected PMH location to operate the distribution system for 6–24 months [40].

The assumptions justified in this study are adapted from [10] as follows:

- PMH locations can be booked every day at a fixed price;
- The delivery zone is served by a maximum of one PMH fleet;
- PMH carries out the process of loading the pick-up package and unloading the package that will be delivered within 30 min;
- PMH's capacity is large enough to accommodate packages that will be delivered to the designated unit zone;
- Delivery officers pick up and deliver packages in one zone within 4 h per stage.

### 2.3. Determination of the Type, Quantity, and Capacity of PMHs

We are modeling the 2E-SS-CVRP-PMH-SPD distribution system using a fleet with a specific capacity that is adjusted according to the density of the demand for parcels to be delivered. In this study, a van/truck with a capacity of 700–1200 kg was used as the PMH. The sleek form of a minibus with various capacities is considered the most suitable for urban conditions in Indonesia and other developing countries. Research has been conducted on 1240 e-buyers from the 6 largest marketplaces in Indonesia regarding reputation, price comparison, and logistics services. Most respondents buy fashion products at a price of IDR 100,000–IDR 500,000 with a weight of 2.5 kg [16].

### 2.4. Determination of Routing and Scheduling Using the 2E-SS-CVRP-PMH-SPD Model

For two-tier systems, tactical planning concerns the departure times, routes, and loads of urban trucks and city freighters, the routing of demand, and the utilization of satellites and the distribution of work among them [25]. Determining the schedule and route begins by classifying the area so that the delivery area is reachable for the deliveryman. The most widely used algorithm for this task is the K-means algorithm, which is known for its simplicity and fast convergence. We counted the number of distribution clusters of packages in 2020 and found that there were 2,037,426 using the silhouette coefficient. This algorithm is appropriate to use with between 1 and 10 million data points [41].

This study uses the saving algorithm and the MILP model to determine the PMH routes and travel schedules. The MILP model ($min \sum_{r \in R} k(r)\rho(r) + \sum_{w \in W} k(w)\varphi(w)$) is the objective function for PMH operational costs and city freighters. It has five constraints, namely (1) PMH capacity ($p$), (2) city freighter capacity ($p$), (3) PMH itinerary ($t,p$), (4) the capacity of PMH locations to provide delivery vehicles ($v$), and (5) the number of city freighters used simultaneously ($n_v$). As for the decision variables, there are 3 variables, namely, (1) PMH work assignment, (2) city freighter work assignment, and (3) itineraries for PMHs and city freighters ($t,p$). An optimization model with the following formula was adapted from [25]:

$$min \sum_{r \in R} k(r)\rho(r) + \sum_{w \in W} k(w)\varphi(w)$$

$$subject\ to\ \sum_{d \in \mathcal{D}} \sum_{m \in \mathcal{M}(d,r)} vol(d)\zeta(m) \leq u_T \rho(r)\ \ r \in \mathcal{R},$$

$$\sum_{d \in \mathcal{D}} \sum_{m \in \mathcal{M}(d,w)} vol(d)\zeta(m) \leq u_v \varphi(w)\ \ w \in \mathcal{W},$$

$$\sum_{m \in \mathcal{M}(d)} \zeta(m) = 1\ \ d \in \mathcal{D},$$

$$\sum_{t^- = t - \delta(T) + 1}^{t} \sum_{w \in \mathcal{W}(s,t^-)} \varphi(w) \leq \lambda_s\ \ s \in \mathcal{S},\ t = 1, \dots, T,$$

$$\sum_{w \in \mathcal{W}(v)} \varphi(w) \leq n_v\ \ v \in \mathcal{V},$$

$$\rho(r) \in \{0,1\}\ \ r \in \mathcal{R},$$

$$\varphi(w) \in \{0,1\}\ \ w \in \mathcal{W},$$

$$\zeta(m) \in \{0,1\} m \in \mathcal{M}(d),\ d \epsilon \mathcal{D},\ \ \ \ \ \ \ \ \ \ (1)$$

$k(r)$ = PMH operational cost from distribution center to parcels and satellites.
$k(w)$ = Operational cost of sending city freighters to deliver or pick up parcels.
$\rho(r)$ = 1 if the urban vehicle service $r \in \mathcal{R}$ is selected (dispatched); otherwise 0.
$\varphi(w)$ = 1 if the city freighter assignment $r \in \mathcal{R}$ is selected (operated); otherwise 0.
$\zeta(m)$ = 1 if the itinerary $m \in \mathcal{M}$ of demand $d \in \mathcal{D}$ is used; otherwise 0.

The terms are explained in detail in Table 2.

**Table 2.** Description of each term.

| Notation | Remark |
|---|---|
| $\mathcal{E} = \{e\}$ | Set of external zones (distribution center). |
| $P = \{p\}$ | Set of products (express service). |
| $\mathcal{C} = \{c\}$ | Set of customers. |
| $\mathcal{D} = \{d\}$ | Set of demand: volume *vol(d)* of product *p(d)* available, starting in period *t(d)* at the external zone *e(d)*, to be delivered to a customer *c(d)* during the time interval (*a(d)*, *b(d)*)); $\delta(d)$: service time at the customer. |
| $\mathcal{T} = \{\tau\}$ | Set of PMH vehicles. |
| $u_\tau$ | Capacity of urban truck type $\tau$. |
| $n_\tau$ | Number of urban trucks of type $\tau$. |
| $\mathcal{T}(p)$ | Set of urban vehicle types that may be used to transport product *p*. |
| $\mathcal{V} = \{v\}$ | Set of city freighter types. |
| $u_v$ | The capacity of city freighter type *v*. |
| $n_v$ | Number of city freighters of type *v*. |
| $\mathcal{V}(p)$ | Set of city freighter types that may be used to transport product *p*. |
| $\mathcal{S} = \{s\}$ | Set of mobile hubs/PMH locations. |
| $\lambda_s$ | The capacity of satellites in terms of the number of city freighters it may accommodate simultaneously. |
| $\delta(\tau)$ | Time required to unload an urban vehicle of type $\tau$ at any satellite. |
| $\delta_{ij}(t)$ | Travel time between two points *i,j* in the city, where each point may be a customer, an external zone, a satellite, or a depot; travel is initiated in period *t*, and the duration is adjusted for the corresponding congestion conditions. |

The number and type of PMH vehicles in the distribution system, as well as the routes and schedules generated from the mathematical model, are used to calculate the operational costs and the resulting emissions. The sustainable performance of various scenarios under the 2E-SS-CVRP-PMH-SPD distribution system is compared with that of the existing distribution system.

## 2.5. Calculation of Sustainable Performance Indicators

Parcel delivery in urban areas involves many parties, including sellers/buyers, carriers/operators, customers, the community, and local authorities or city governments. Each party has different roles and interests [12]. The roles and interests of stakeholders involved in e-commerce parcels in urban areas are described in Table 3.

The distribution of parcels in urban areas and the flow of private vehicles are significant sources of energy consumption, air pollution, and noise. As a result, in the 2000s, several cities in the European Union began implementing sustainable urban freight solutions. In this context, this research aims to discuss the application of a new and more ecological freight transportation system that serves several adjacent urban areas using environmentally friendly vehicles to deliver goods [42].

**Table 3.** The roles and interests of stakeholders. Adapted from [12].

| Stakeholder | Role | Interest |
|---|---|---|
| CEP operators | • Operationalization of PMH<br>• Parcel delivery and pickup | • Provide as much service possible at the lowest cost possible (economics)<br>• The deliveryman's work is easier and more effective (social) |
| Shippers | • Send the parcel (online)<br>• Pay the parcel fee (companies/MSME/individuals) | Better services (fast) with flat cost (social) |
| Buyers | Receive the parcel | Better services (fast) with a flat cost even if practical operations are changed |
| Citizens | People who live, work, and spend their free time in the city | A safe and healthy environment |
| Local authorities | Municipality | Increase the livability of the city in many ways (pollution, safety, and congestion) |

There are various key performance indicators (KPIs) when delivering a parcel. Albergaria et al. [43] summarized the expected benefits of establishing a parcel delivery model. The benefits consist of economic (delivery time, operational costs, energy consumption, congestion, etc.), environmental (reduction of $CO_2$ emissions, noise, air pollutants, etc.), and social (quality of life, job creation, etc.) aspects.

Developing a sustainable transportation system in urban or metropolitan areas comes with enormous challenges. The transportation profile in metropolitan regions in Indonesia is based on the assessment of fourteen indicators of sustainable transportation applied in this study. Still, it has not yet been shown that metropolitan areas in Indonesia have the potential to develop sustainable transportation [44].

According to [45], the priorities of e-buyers are (1) the price of goods, (2) the variety of types of goods, (3) the convenience of transactions, and (4) speed. Other studies that have looked at the possible implications of e-commerce have analyzed broader structural changes and their potential to make the delivery of goods more efficient [39].

Morganti et al. [46] cite the research of [47] and state that most problems related to online shopping are related to delivery rather than the product itself. The study notes that 39% of e-consumers experience problems such as home delivery when no one is home/failed to deliver (15%), delivery delays (13%), excessive delivery costs (7%), incomplete tracking of delivery status (5%), and the need to collect products from remote collection points (3%). There is a slightly different e-commerce problem in Germany. In particular, consumers report negative experiences with late delivery (29%) and damaged parcels (20%) [48].

Meanwhile, [27] aimed to build a framework that could help last-mile operators form an efficient logistics organization that fits the city's characteristics via an appropriate and complete regional diagnosis.

We use the economic aspect of performance to minimize the operational costs of a two-echelon distribution system involving PMHs to deliver parcels to customers. Using the abovementioned formula, a PMH's operating costs can be explained as follows:

1. PMH operations consist of the total cost of the PMH fleet, the cost of the city freighter fleet in the form of motorbikes, the cost of the PMH driver, and the operating costs of the deliveryman.
2. The PMH fleet cost (PFC) consists of:
   a. The PMH fleet fee;
   b. The PMH fleet operational costs, including fuel, parking, and tolls;
   c. The PMH driver cost (PDC) consists of the drivers' salaries/incentives;
3. The operational cost of deliverymen (OCD) consists of:

      a.      The operational cost of the motorcycles, especially fuel;

      b.      The deliverymen's salaries/incentives;

4.    The total cost of a PMH (TCP):

TCP (IDR/parcel) = (PFC + PDC + OCD): number of parcels delivered/picked up.

Operationally, the delivery time per parcel indicates the efficiency of the workers operating within the system. It can be defined as the vehicle's total travel time divided by the number of parcels handled. Specifically, a deliveryman's productivity is the average time between customer sites on the delivery route. That is, the ratio of the deliveryman's operating time to the number of parcels handled. Knowing the deliveryman's opinion is critical to the successful use of a PMH. The deliveryman's productivity increases if he/she is paid depending on the number of parcels picked up or delivered [10]. In this study, we measure the social aspect of the acceptance of the PMH concept from the operator's side (the deliveryman). The acceptance of the PMH concept is very significant for the success of field implementation.

The environmental footprint of PMH operations consists of direct impacts and negative externalities. Negative externalities include congestion caused by vehicle movement and parcel pickup and delivery stops. A standard measurement for assessing direct environmental impacts is $CO_2$ emissions from vehicle movement, calculated as the total distance traveled times the emission factor for each vehicle type. Rider vehicles are typically more prominent (e.g., parcel cars, delivery vans), faster, and emit more pollutants than delivery vehicles, which are designed to be convenient and respectful in dense city centers (e.g., tricycles or electric motorcycles) [10]. We calculated the $CO_2$ emissions using the criteria in [49] with the following formula:

$$Emission = \sum\nolimits_a [Fuel_a \; x \; EF_a] \qquad (2)$$

$Emission$ = $CO_2$ emissions (kg)
$Fuel_a$ = fuel usage (terra joule/TJ)
$EF_a$ = emission factor (kg/TJ). This value is equal to the carbon content of the fuel multiplied by 44/12
$a$ = fuel type (e.g., premium, diesel, natural gas, LPG, etc.)

### 2.6. Case Study Description

The conceptual framework of PMH sites was used to construct the 2E-SS-CVRP-PMH-SPD distribution model. The city of Bandung is used as a case study to test this distribution model.

Bandung is a city in Indonesia (located between 107° East Longitude and 6°55′ South Latitude) that represents a city in a developing country [50]. Bandung is used as a synthetic or illustrative case study because it represents cities in Indonesia in online trade transactions [51] with the toughest transportation sustainability challenges among the 6 biggest cities in Indonesia, with a score of 3 out of 14 [44].

### 2.7. Research Data Sources

This study performs a spatial analysis to determine the best location for a PMH as part of an e-commerce parcel distribution system in Bandung while considering three main variables and two constraints. This case study is expected to illustrate and synthesize PMH location selection in an electronic trading parcel distribution system with sustainable performance indicators.

The data in this study come from various sources, with e-commerce parcel data sent through a courier operator from January to December 2020 being the primary data. The data on quota usage for online purchase transactions comes from one mobile phone operator in Indonesia, and the location of e-buyer centers (offices, student dormitories, and apartments) was captured from Google Maps.

## 3. Result and Discussion

### 3.1. Data Collection and Processing

Bandung city consists of 30 sub-districts (Figure 3a) and 151 urban villages (Figure 3b). The primary data in this research are related to e-commerce parcels in Bandung city between January and December 2020, which were sent through a courier operator (CEP operator). The number of e-commerce parcels delivered by the courier operator in Bandung in 2020 was 2,037,426, with an average of 5660 parcels/day being delivered, as is shown in Figure 4.

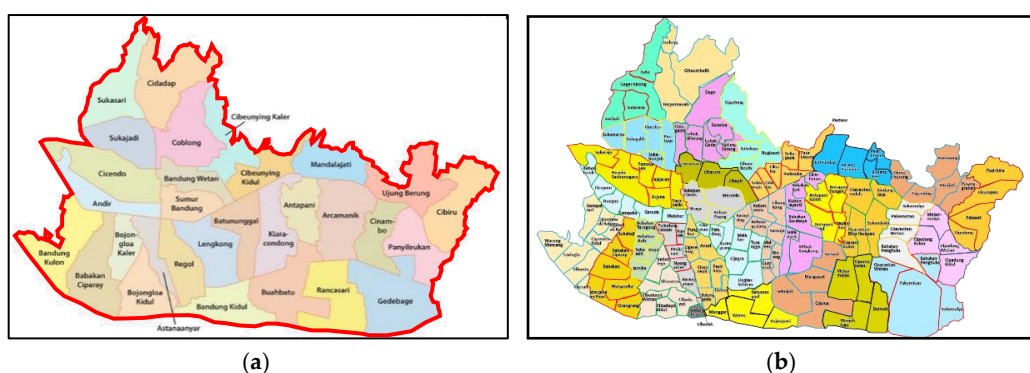

| (**a**) | (**b**) |

**Figure 3.** Bandung city map. (**a**) Sub-districts. (**b**) Urban villages.

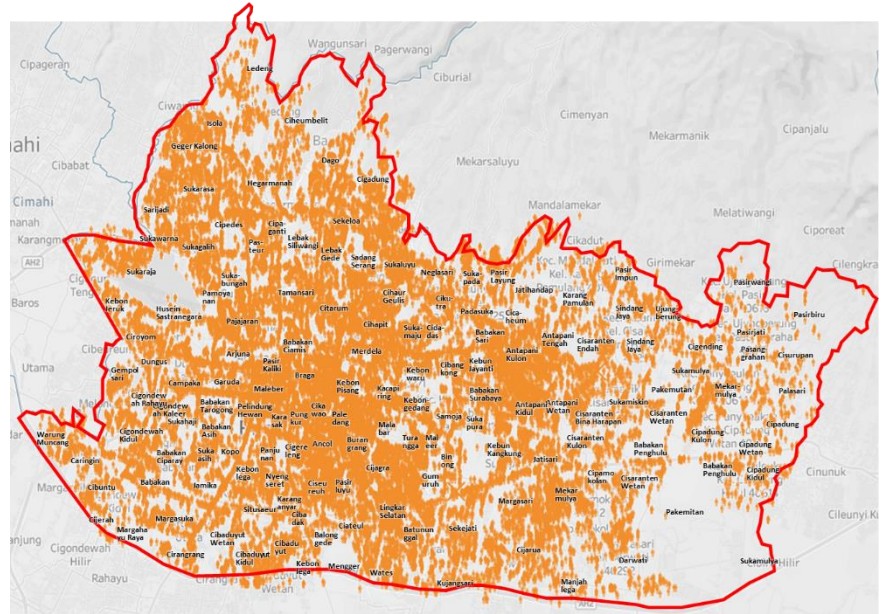

**Figure 4.** E-commerce parcel distribution in Bandung between January and December 2020.

The PMH's location is the center of gravity (CoG), which combines the number of parcels requested (parcels demand), quota usage for e-commerce, and e-buyer bags. The source of CoG data for parcel demand comes from the courier operator, the quota usage data for e-commerce are based on a cellular operator in Indonesia, and the data on buyers comes from the spatial data of apartments, student dormitories, and offices captured from Google Maps.

Furthermore, we collected traffic availability data for the PMH location candidates. The traffic data around the PMH location candidates were collected by measuring traffic flow on two-way roads during the day from 09.00 to 15.00 WIB from 7–21 May 2021, when PPKM Level 3 is applied, and traffic conditions are about 50% of their usual level.

The parcel distribution data were displayed using Tableau 2018 2.0. CoG calculations, schedule determination, and PMH routes with the saving and optimization algorithm were processed using Python 3.9.

### 3.2. Determination of the Number of Clusters

The distribution of clusters in Bandung using the K-means method produces silhouette coefficient values, which are shown in Table 4.

**Table 4.** Silhouette coefficients using the K-means method.

| Number of Clusters | Silhouette Coefficient Value | Rank |
|---|---|---|
| 1 | - | - |
| 2 | 0.45582675319963720 | 1 |
| 3 | 0.36520875766877960 | 9 |
| 4 | 0.38307243667650710 | 8 |
| 5 | 0.39287226705658157 | 7 |
| 6 | 0.41702320039842740 | 2 |
| 7 | 0.41599444461142326 | 4 |
| 8 | 0.41040660200445080 | 5 |
| 9 | 0.41600486734332126 | 3 |
| 10 | 0.40774175075599070 | 6 |

The three highest silhouette coefficient values were found for clusters two (0.456), six (0.417), and nine (0.416). The area of Bandung city is 167.3 km$^2$, so if it is divided into two clusters, the delivery area is still very wide. Therefore, it is possible to use six clusters (Figure 5a) or nine clusters (Figure 5b).

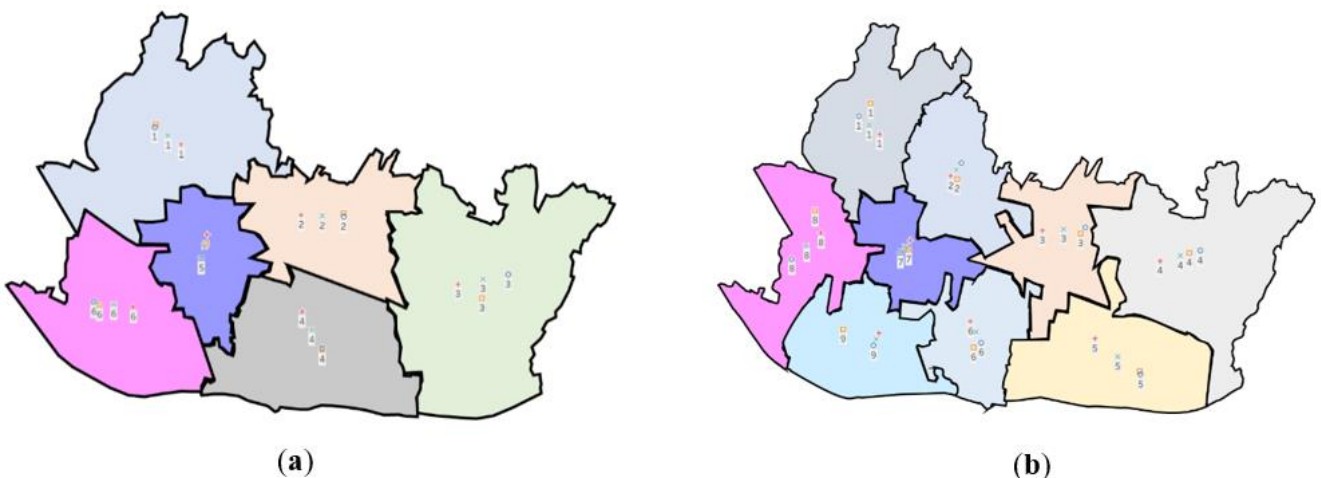

(a)     (b)

**Figure 5.** CoG location of parcel, e-buyer, and quota using (**a**) six clusters and (**b**) nine clusters.

### 3.3. Relationship between Variables

Table 5 shows the location of the parcels' CoG (columns 1–2), centers of e-buyers CoG (columns 3–4), quota usage CoG (columns 5–6), and the average e-buyer and quota usage CoG (columns 7–8). Column 9 is the distance between the parcel and the e-buyer's CoG, while column 10 is the distance between the parcel and the quota usage's CoG. Column 11 is the distance between the parcel and the average e-buyer with quota usage's CoG. The best PMH location is the parcel CoG because it is the closest to the customer. However, it is not easy to obtain parcel request data, and it is not always available. Therefore, parcel CoG can be represented by the center of the e-buyer CoG, the quota usage CoG, or the average of both.

Based on the distribution of parcels in 2020, the quota usage, and the centers of e-buyers, the center of gravity (CoG) for each cluster is determined. The results are presented in Table 6 for six clusters and Table 7 for nine clusters.

**Table 5.** CoG based on parcels, e-buyers, quotas, and distance in six clusters.

| Cluster | Parcel/X1 | | E-Buyer/X2 | | Quota/X3 | | Mean/X4 = (X2 + X3)/2 | | Distance (km) | | | Outermost Point Distance (km) | Distance/Outermost Point Distance | | |
|---|---|---|---|---|---|---|---|---|---|---|---|---|---|---|---|
| | Lat | Long | Lat | Long | Lat | Long | Lat | Long | X1–X2 | X1–X3 | X1–X4 | | X1–X2 | X1–X3 | X1–X4 |
| | (1) | (2) | (3) | (4) | (5) | (6) | (7) | (8) | (9) | (10) | (11) | (12) | (13) | (14) | (15) |
| 1 | −6.88149 | 107.59554 | −6.88848 | 107.59423 | −6.88261 | 107.59521 | −6.88555 | 107.59472 | 0.79 | 0.13 | 0.46 | 8.65 | 9.1% | 1.5% | 5.3% |
| 2 | −6.90741 | 107.64614 | −6.90822 | 107.63973 | −6.90880 | 107.64620 | −6.90851 | 107.64296 | 0.71 | 0.15 | 0.37 | 7.94 | 9.0% | 1.9% | 4.7% |
| 3 | −6.93206 | 107.68941 | −6.92728 | 107.68128 | −6.92395 | 107.69843 | −6.92562 | 107.68986 | 1.04 | 1.34 | 0.72 | 9.72 | 10.7% | 13.8% | 7.4% |
| 4 | −6.94468 | 107.64150 | −6.94184 | 107.63453 | −6.94424 | 107.64146 | −6.94304 | 107.63799 | 0.83 | 0.05 | 0.43 | 6.97 | 11.9% | 0.7% | 6.1% |
| 5 | −6.91693 | 107.61152 | −6.91386 | 107.61193 | −6.92190 | 107.60985 | −6.91788 | 107.61089 | 0.34 | 0.58 | 0.13 | 6.71 | 5.1% | 8.7% | 1.9% |
| 6 | −6.93554 | 107.58367 | −6.93621 | 107.59111 | −6.93451 | 107.58173 | −6.93536 | 107.58642 | 0.82 | 0.24 | 0.30 | 5.23 | 15.8% | 4.6% | 5.8% |
| Mean | | | | | | | | | 0.76 | 0.42 | 0.40 | 7.54 | 10.3% | 5.2% | 5.2% |
| SD | | | | | | | | | 0.18 | 0.40 | 0.16 | 1.58 | | | |

**Table 6.** CoG based on parcels, e-buyers, quotas, and distance in nine clusters.

| Cluster | Parcel/X1 | | E-Buyer/X2 | | Quota/X3 | | Mean/X4 = (X2 + X3)/2 | | Distance (km) | | | Outermost Point Distance (km) | Distance/Outermost Point Distance | | |
|---|---|---|---|---|---|---|---|---|---|---|---|---|---|---|---|
| | Lat | Long | Lat | Long | Lat | Long | Lat | Long | X1–X2 | X1–X3 | X1–X4 | | X1–X2 | X1–X3 | X1–X4 |
| | (1) | (2) | (3) | (4) | (5) | (6) | (7) | (8) | (9) | (10) | (11) | (12) | (13) | (14) | (15) |
| 1 | −6.87343 | 107.59547 | −6.87654 | 107.58981 | −6.87869 | 107.59069 | −6.87761 | 107.59025 | 0.71 | 0.79 | 0.74 | 7.15 | 10.0% | 11.0% | 10.4% |
| 2 | −6.89967 | 107.62041 | −6.89934 | 107.61870 | −6.89304 | 107.62218 | −6.89619 | 107.62044 | 0.19 | 0.76 | 0.39 | 6.98 | 2.8% | 10.9% | 5.5% |
| 3 | −6.91220 | 107.65145 | −6.91085 | 107.64272 | −6.90968 | 107.65328 | −6.91026 | 107.64800 | 0.98 | 0.35 | 0.44 | 10.25 | 9.5% | 3.4% | 4.3% |
| 4 | −6.92201 | 107.69871 | −6.92537 | 107.69277 | −6.92119 | 107.70331 | −6.92328 | 107.69804 | 0.75 | 0.52 | 0.16 | 5.52 | 13.7% | 9.3% | 2.9% |
| 5 | −6.94853 | 107.66494 | −6.93935 | 107.66654 | −6.94993 | 107.66521 | −6.94464 | 107.66587 | 1.04 | 0.16 | 0.44 | 7.68 | 13.5% | 2.1% | 5.8% |
| 6 | −6.94124 | 107.62593 | −6.93467 | 107.62568 | −6.93929 | 107.62906 | −6.93698 | 107.62737 | 0.73 | 0.41 | 0.50 | 7.41 | 9.9% | 5.5% | 6.7% |
| 7 | −6.91944 | 107.60930 | −6.91564 | 107.61008 | −6.92061 | 107.60601 | −6.91812 | 107.60805 | 0.43 | 0.39 | 0.20 | 5.83 | 7.4% | 6.6% | 3.5% |
| 8 | −6.90318 | 107.58376 | −6.90624 | 107.58313 | −6.90289 | 107.57408 | −6.90456 | 107.57860 | 0.35 | 1.07 | 0.59 | 7.00 | 5.0% | 15.3% | 8.4% |
| 9 | −6.93994 | 107.58116 | −6.94036 | 107.58974 | −6.94042 | 107.59059 | −6.94039 | 107.59017 | 0.95 | 1.04 | 1.00 | 4.58 | 20.7% | 22.8% | 21.8% |
| Mean | | | | | | | | | 0.68 | 0.61 | 0.49 | 6.93 | 10.3% | 9.7% | 7.7% |
| SD | | | | | | | | | 0.28 | 0.30 | 0.24 | 1.61 | | | |

**Table 7.** Candidate and selected locations for the PMH fleet.

| Cluster | Address | Candidate Locations for PMHs | | | | | | Selected Location |
|---|---|---|---|---|---|---|---|---|
| | | Candidate#1 | | Candidate#2 | | Candidate#3 | | |
| | | Lang/Lot | Flow (pce/Hour) | Lang/Lot | Flow (pce/Hour) | Lang/Lot | Flow (pce/Hour) | |
| (1) | (2) | (3) | (4) | (5) | (6) | (7) | (8) | (9) |
| 1 | Tendean St. no. 38 Hegarmanah | −6.873126 107.598116 | 71 | −6.871941 107.600321 | 305 | −6.874613 107.599835 | 213 | Tendean St. no. 39 Hegarmahah (#1) |
| 2 | Melania St. no. 15 Cihaur Geulis | −6.900489 107.626696 | 925 | −6.899461 107.625825 | 3.314 | −6.8991169 107.6267182 | 3314 | Suropati St. no. 104 (#2) |
| 3 | Kalijati Indah Baru St. 14-8 Antapani Kulon | −6.908864 107.654203 | 252 | −6.908531 107.65458 | 252 | −6.908398 107.65396 | 252 | Kalijati Indah Baru St. no. 14 (#3) |
| 4 | Pasanggrahan III St. no. 10 Cipadung Kulon | −6.923303 107.705483 | 203 | −6.922679 107.704337 | 203 | −6.922415 107.704537 | 203 | Pasanggrahan Raya St. (#2) |
| 5 | Wastukencana St. no. 3 Babakan Ciamis | −6.907315 107.58482 | 2950 | −6.907102 107.585931 | 318 | −6.905962 107.579648 | 3022 | Nurtanio Utara St. (#2) |
| 6 | Pungkur St. no. 231 Balonggede Regol | −6.928016 107.609071 | 1730 | −6.926118 107.608782 | 517 | −6.925912 107.608782 | 517 | Pasundan St. no. 66 (#2) |
| 7 | Amd 9 St. no. 141 Babakan Tarogong Bojongloa Kaler | −6.941117 107.583671 | 825 | −6.940188 107.581647 | 3.822 | −6.938662 107.5828548 | 825 | Babakan Ciparay St. no. 223 (#3) |
| 8 | Karapitan St. no. 57A | −6.942467 107.631486 | 46 | −6.941988 107.631775 | 46 | −6.941727 107.631831 | 46 | Rajamantri Kulon St. no. 17 (#1) |
| 9 | Babakan Jati St. no. 128 Gumuruh | −6.9461882 107.6663679 | 235 | −6.9477699 107.6659526 | 462 | −6.9469625 107.6651699 | 154 | Mars Raya St. no. 1 (#3) |

The quota CoG (X3) can represent the parcel CoG (X1) with an average distance of 0.61 km and a standard deviation of 0.3 km, which is better than the e-buyer CoG (X2), which has an average length of 0.68 km and a standard deviation of 0.28 km. Meanwhile, the quota and e-buyer CoG (X4) has an average distance of 0.49 km and a standard deviation of 0.24 km. Compared to the distance to the outermost point, the distance between these variables has a value of 10% (X2) and 5% (X3 and X4). This value of less than 10% indicates that the quota, e-buyers, and average of these two variables can represent X1.

For the same reason, Table 6 shows that quota CoG (X3) can represent parcel CoG (X1, with an average distance of 0.42 km and a standard deviation of 0.4 km) better than e-buyer CoG (X2, with an average distance of 0.76 km and a standard deviation of 0.18 km). Meanwhile, the average quota and e-buyer CoG (X4) has an average distance of 0.40 km and a standard deviation of 0.16 km. The distances between these variables, when compared with the distance to the outermost point, have values of 10% (X2), 10% (X3), and 8% (X4). Values below 10% indicate that the quota variable, e-buyers, and the average of these two variables can represent X1. In other words, the conceptual framework for choosing PMH locations (Figure 3) is influenced by parcel distribution, quota usage, and the distribution of e-buyer centers.

### 3.4. Determination of PMH Locations and Clustering

PMH location candidates are determined based on direct survey data from the field that assesses the availability of land and traffic conditions in the area around the CoG with a diameter of 500 m. The selected PMH location has a minimum available land area of 200 m$^2$ (to accommodate the PMH fleet with approximately 12 motorbikes) with access to the best location (indicated by the most minor traffic flow). Table 7 summarizes the survey results, and column 2 is the CoG address of each cluster. Meanwhile, columns 3 to 8 are candidates for PMH locations around the CoG with a diameter of 500 m, with land above 200 m$^2$. Column 9 is the selected location with access to the best site.

### 3.5. Calculation of PMH Total Capacity (Base on Parcel's CoG)

The PMH used is a van/truck with a capacity of 700–1200 kg, an ideal vehicle for transporting parcels and moving agilely on urban roads in Indonesia. The different PMH transportation modes include a Grandmax Blindvan (GB; capacity 0.72 tons), a Grandmax 1.3 L (G 1.3 L; capacity 1 ton), and a Grandmax 1.5 L (G 1.5 L; capacity 1.1 tons). The number of daily parcels was calculated based on the total in January 2020, which was 124,561 tons (228,472 items).

### 3.6. Routing and Scheduling

Calculating PMH routes and schedules using the Clark and Wright/saving algorithm proposed seven alternatives, namely one existing condition and six alternatives based on the saving algorithm and optimization in clusters six and nine, as are shown in Appendix B (Tables A4–A10). The calculation of the PMH routes and schedules is summarized in Table 8.

**Table 8.** PMH fleet profile used from alternative 0 to 6.

| Alternative | Number of Clusters | Method | Number of PMH Fleets | Average Load Factor |
|---|---|---|---|---|
| 0 | Existing | - | Two fleets (2 G 1.5 Ls) | 94% |
| 1 | 6 | Saving Algorithm | Three fleets (2 GBs, 1 G 1.3 L) | 86% |
| 2 | 6 | Saving Algorithm | Two fleets (2 G 1.5 Ls) | 94% |
| 3 | 6 | Optimization | Three fleets (2 GBs, 1 G 1.3 L) | 94% |
| 4 | 9 | Saving Algorithm | Five fleets (5 GBs) | 86% |
| 5 | 9 | Saving Algorithm | Three fleets (2 GBs, 1 G 1.3 L) | 58% |
| 6 | 9 | Optimization | Three fleets (3 GBs) | 86% |

From Table 8, we can see that the lowest number of fleets used is two in alternatives 1 and 3, while the greatest number used is five in alternative 5, with an average load factor of 86%. The highest load factor is achieved with alternatives 1, 3, and 4 (94% each).

### 3.7. Distribution Model Validation

Model validation ensures that the model can make accurate predictions in site-specific field settings. Successful validation requires the completion of steps that make up the modeling protocol, including model design and calibration and the verification of the equations, computer code, and model itself. The most rigorous validation demonstrates that the model can accurately predict the future (Anderson and Woessner, 1992).

The two-echelon distribution model of e-commerce parcels generated using data on the distribution of parcels in 2020 was validated using data from 2021.

The e-commerce parcel distribution model in which the PMH location is determined based on the 2020 parcel distribution data is validated with the 2021 data with the following results:

- The distance of the outermost point of each cluster is between 4.58 km to 10.25 km, with an average of 6.93 km;
- The parcel distribution CoG in 2020 vs. 2021 has a distance of between 0.27 km and 1.81 km with an average of 0.74 km (column 10) or 9.8% of the average length of the outermost point of each cluster.

By using a distance tolerance of 10% ($\alpha$), the calculation result of the parcel distance CoG distribution is 9.8%, proving that the e-commerce parcel distribution model in which PMH locations are determined based on the parcel distribution CoG in 2020 is validated by the distribution of parcels in 2021.

### 3.8. Calculation of Sustainable Transport Indicators

The calculation of sustainable transportation indicators against the distribution model 2E-SS-CVRP-PMH-SPD was carried out by comparing the existing conditions (scenario A) with three other scenarios (scenarios B, C, and D) where the city freighters are conventional motorcycles, electric motorcycles, and electric scooters. The results of the calculation of sustainable transportation indicators from various scenarios and alternatives can be seen in Table 9.

**Table 9.** The calculation result of economic and environmental indicators for the 2E-SS-CVRP-PMH-SPD distribution system with various scenarios and alternatives.

| Scenarios and Alternatives | Economy (IDR/Item) | | Environment/$CO_2$ (kg/Year) |
|---|---|---|---|
| | Delivery | SPD | |
| (1) | (2) | (3) | (4) |
| Scenario A | 3392 | 3084 | 413.95 |
| Scenario B Alt-1 | 3216 | 2392 | 222.69 |
| Scenario B Alt-2 | 3188 | 2512 | 223.18 |
| Scenario B Alt-3 | 3201 | 2462 | 202.51 |
| Scenario B Alt-4 | 3295 | 2602 | 235.42 |
| Scenario B Alt-5 | 3221 | 2478 | 229.04 |
| Scenario B Alt-6 | 3218 | 2475 | 227.88 |
| Scenario C | 4604 | 3542 | 19.31 |
| Scenario D | 3613 | 2779 | 19.31 |

The calculation of the economic indicator yield cost per parcel was conducted under two scenarios: (1) delivery only (column 2) and (2) simultaneous pickup and delivery

(column 3). The calculation of environmental indicators by measuring the production of $CO_2$ is shown in column 4.

Social indicators measure the acceptance of the PMH concept from the operator's perspective (deliveryman). Interviews were conducted with 30 deliverymen from the courier operator on 30 March 2021. Almost all interviewees expressed their agreement with the PMH concept with three main caveats: they wished to be provided with working facilities at the PMH location (sorting table), the PMH should be equipped with a large umbrella to protect parcels from the sun and rain, and the incentive scheme should be more attractive. There were two interviewees (12.5%) who expressed objections to the PMH concept because their house happened to be closer to the existing DC compared to the PMH location.

The ordering of economic indicators by delivery cost per parcel (Appendix C, Table A11, column 2) shows that alternative B scenario 2 is the best scenario with the lowest cost of DIR 3188 per parcel for six clusters. Meanwhile, for nine clusters, the best scenario is scenario B, alternative 6, which has a cost of IDR 3218 per parcel. The order of economic indicators by simultaneous pickup and delivery cost per parcel (Appendix C, Table A11, column 3) shows that alternative B scenario 1 is the best scenario with the lowest cost of IDR 2393 per parcel for six clusters (down 22.4% compared to scenario A). Meanwhile, for nine clusters, the best scenario is scenario B, alternative 6, with a cost of IDR 2475 per parcel. The order of environmental indicators by $CO_2$ production (Appendix C, Table A11, column 4) shows that scenarios C and D have the lowest $CO_2$ production of 19.3 kg per year. For six clusters, the best scenario is scenario B, alternative 3, with a $CO_2$ production of 202.51 kg per year. For nine clusters, the best scenario is scenario B, alternative 6, with a $CO_2$ production of 227.88 kg per year.

The use of electric motorcycles or scooters as city freighters could reduce $CO_2$ emissions by 95% compared to the existing conditions. However, operational costs would increase by 7% (scooters) and 36% (electric motorbikes) due to the high cost of renting electric vehicles. In the context of Indonesia and other developing countries, the use of electric vehicles (scenarios C and D) is not an option over the next 5 years due to a lack of infrastructure for charging and replacing batteries.

Therefore, one possible alternative is using scenario B alternative 6, which is the best economic and environmental alternative for dividing Bandung city into nine clusters. This scenario causes a decrease in operational costs from 3084 IDR/item to 2475 IDR/item (19.7%; Table A11, SPD cost/**) and decreases $CO_2$ production by 44.9% (down from 413.95 kg/year to 227.88 kg/year). This scenario reduces $CO_2$ emissions by 186 kg/year. The decrease in operational costs and $CO_2$ emissions is due to the reduced distance traveled by PMHs and motorbikes compared to existing conditions.

There are 1350 deliverymen working for the courier operator (about 5.5% of CEP operators) [52]. Assuming that all CEP operators use the PMH concept, the prevention of additional $CO_2$ in Bandung is 186 kg/year times (100%/5.5%) = 3.445 kg/year or 3.4 tons per year. Assuming that the electricity ecosystem is well developed, the scenarios chosen are scenario C (city freighter in the form of a scooter) and scenario D (city freighter in the form of an electric motorcycle). This scenario results in a distribution system that can prevent the addition of 7.2 tons of $CO_2$ per year.

### 3.9. Results

This study proposes an innovative distribution system with PMH as a conceptual framework for selecting PMH locations for e-commerce parcels. The conceptual framework for selecting PMH locations is influenced by three variables and two constraints. Testing the conceptual framework through spatial analysis using package distribution data for a courier operator in 2020 in two stages.

The first stage examines the relationship between variables that influence the selection of PMH locations. The test results show that the relationship between variables is significantly correlated.

The second stage tested the innovative distribution system with better PMH compared to the classic distribution system using sustainable performance indicators. Economic indicators show more efficient distribution costs, with a cost of 19.7%. Social indicators are measured through in-depth interviews with operator officers with the results of acceptance of the PMH concept. The environmental indicators are measured by calculating $CO_2$ production with a smaller yield of 44.9%.

The conceptual framework for selecting PMH locations for e-commerce parcels can be used in other cities where the CoG of package distribution is influenced by the CoG of quota usage and the center of e-buyers. The center of e-buyers for each city may be different. In this research, the center of e-buyers uses CoG from offices, student dormitories, and apartments. In another city, the center of e-buyers can be a shopping center, residential center, education center, etc.

The three variables and two constraints tested in this study are an extension of previous research.

## 4. Conclusions and Next Research

The 2E-SS-CVRP-PMH-SPD distribution system is an innovative distribution system that combines parcel mobile hubs (PMHs) with small vehicles such as motorcycles. It has been proven to increase operational cost-efficiency. This research can provide the conceptual framework for selecting PMH locations using a spatial approach. The best PMH locations are determined by the parcel demand CoG, which is influenced by the quota usage CoG and the e-buyer center CoG with the constraints of space availability and traffic flow conditions in the vicinity.

This study proposes a new conceptual framework in the form of selecting PMH locations by taking into account three variables and two constraints, not just relying on the center of the zone. The three variables are package distribution, internet quota usage for e-commerce, and the center of e-buyers. The two constraints consist of space availability and traffic flow conditions around the location.

The development of this innovative distribution system model can produce better sustainable transportation performance compared to the classic distribution system that uses a physical building as a hub. Using a PMH combined with conventional motorcycles can reduce operational costs by 19.7% and prevent potential $CO_2$ emissions by 44.9%. The use of the PMH concept by all operators in Indonesia is projected to prevent potential $CO_2$ production by up to 3.4 tons per year. If the city freighter is an electric motor or scooter, it can further prevent potential $CO_2$ production by up to 7.2 tons per year.

In further research, e-buyer centers can be studied with data on the number of occupants in each office, student dormitory, and apartment to help produce a more accurate distribution model.

**Author Contributions:** Conceptualization, R.P.; Methodology, R.P.; Software, R.P.; Validation, G.Y. and P.D.; Formal analysis, R.P.; Investigation, R.P.; Resources, R.P.; Data curation, R.P.; Writing—original draft, G.Y. and P.D.; Writing—review & editing, H.P.H.P., G.Y. and P.D.; Supervision, H.P.H.P., G.Y. and P.D.; Project administration, G.Y. All authors have read and agreed to the published version of the manuscript.

**Funding:** This research received no external funding.

**Informed Consent Statement:** Informed consent was obtained from all subjects involved in the study.

**Data Availability Statement:** Not applicable; this study does not report any data.

**Conflicts of Interest:** The authors declare no conflict of interest.

## Appendix A

**Table A1.** Reviewed papers about mobile access hubs (MAHs) #1.

| Citation | | Del Pia and Filippi, 2006 [53] | Arvidsson and Pazirandeh, 2017 [29] | Marujo et al., 2018 [11] | Moshref-Javadi et al., 2020 [35] |
|---|---|---|---|---|---|
| | | **1** | **2** | **3** | **4** |
| Country | | Italy | Sweden | Brazil (Rio de Janero) | USA |
| Vehicle | Hub | Truck | Bus, truck, barge, tram | Conventional truck | Conventional truck |
| | Satellite | Small vehicle | LEV/cargo cycles | Cargo tricycles | Unmanned Aerial Vehicle (UAV)/drone |
| Commodity | | Garbage | General | General | Social subsidies |
| Location Choice | | n.a. | n.a. | n.a. | n.a. |
| MAH | Number | n.a. | n.a. | n.a. | n.a. |
| | Capacity | n.a. | n.a. | n.a. | n.a. |
| Modeling | Type | Capacitated Arc Routing Problem with Mobile Depots (CARP-MD) | n.a. | Monte Carlo simulation | Simultaneous Traveling Repairman Problem with Drones (STRPD) |
| | Fix | MILP | n.a. | n.a. | MILP |
| | Heuristic | Rendezvous and VND-CARP | n.a. | n.a. | Adaptive Large Neighborhood Search (ALNS) with 8 Algorithms |
| Performance | Economy | n.a. | Cost of standing, running, and overhead | The level of service and the operational cost of the delivery process | Cost objective |
| | Social | n.a. | n.a. | n.a. | Customer waiting time objective |
| | Environment | n.a. | $CO_2$ (Cefic and ECTA, 2011) | GHG, CO, NOx, NMHC, particulate matter (PM), and $CO_2$ | n.a. |

**Table A2.** Reviewed papers about Mobile access hubs (MAHs) #2.

| Citation | | Ding et al., 2019 [18] | Verlinde et al., 2014 [12] | Savuran and Karakaya, 2016 [32] | Bashiri et al., 2018 [36] |
|---|---|---|---|---|---|
| | | **5** | **6** | **7** | **8** |
| Country | | China | Belgium | Germany (Berlin) | Australia |
| Vehicle | Hub | Battery charger robot | Trailer | Conventional truck | Conventional truck |
| | Satellite | Robot | Electric cyclo cargo | Unmanned air vehicle/drone | n.a. |
| Commodity | | Battery | Parcel | General | Services (postal, hospital, etc.) |
| Location Choice | | n.a. | Center of Brussels | n.a. | Dynamic |
| MAH | Number | Simulation | 1 | n.a. | P-mobile |
| | Capacity | n.a. | Trailer (20') | n.a. | n.a. |

**Table A2.** *Cont.*

| Citation | | Ding et al., 2019 [18] | Verlinde et al., 2014 [12] | Savuran and Karakaya, 2016 [32] | Bashiri et al., 2018 [36] |
|---|---|---|---|---|---|
| Modeling | Type | Generalized Multiple Depots TSP (GMDTSP), | (Capacitated, simultaneous pickup & delivery) VRP | Capacitated Mobile Depot VRP (C-MoDVRP) | Dynamic facility location problem |
| | Fix | n.a. | n.a. | n.a. | n.a. |
| | Heuristic | Multiple Depots Random Select (MDRS), dec-MDRS, and MDRS-IM | n.a. | (GA-C-MoDVRP) compared with NN and HC | Genetic Algorithm |
| Performance | Economy | Minimize waiting time | Transport impact/Lead time (95–>87%) | Performance of GA, 2-opt local search method | n.a. |
| | Social | n.a. | Survey of 12 person | n.a. | n.a. |
| | Environment | n.a. | STREAM emission factors (C02, S02, Nox, PM) | n.a. | n.a. |

**Table A3.** Reviewed papers about mobile access hubs (MAHs) #3.

| Citation | | Jeong and Lee, 2019 [31] | Anagnostopoulos et al., 2015 [30] | Leyerer et al., 2019 [34] | Faugère et al., 2020 [10] | Prastyantoro et al., 2022 |
|---|---|---|---|---|---|---|
| | | 9 | 10 | 11 | 12 | 13 |
| Country | | USA | Rusia | Germany | USA | Indonesia |
| Vehicle | Hub | Conventional truck | Big truck | Truck, van, etc. | Trailer | L300/minibus/CDE |
| | Satellite | Drone | Small truck | Van, cargo tricycle, etc. | Cargo tricycle | Motorcycle (electric/conventional), scooter |
| Commodity | | General | Cesspit | Parcel | Parcel | E-commerce parcel |
| Location Choice | | n.a. | n.a. | Decision support system (DSS) | Center of gravity (zone) | CoG from 3 variables and 2 constraints |
| MAH | Number | 1 or 2 | 1 small truck; big truck | n.a. | n.a. | Parcel volume/ PMH capacity |
| | Capacity | n.a. | Small; Big | n.a. | simulation | 720 kg/1000 kg/ 1100 kg |
| Modeling | Type | Vehicle-carrier routing problem (VCRP) | Dynamic VRP | Dynamic VRP | (Capacitated, simultaneous pickup and delivery) VRP | (Capacitated PMH, SS, and SPD) VRP |
| | Fix | Optimation (time and cost) | n.a. | MILP | MILP (cost minimization) | MILP (cost minimization) |
| | Heuristic | n.a. | n.a. | n.a. | n.a. | Clark and Wright (saving algorithm) |
| Performance | Economy | Cost minimization | Cost minimization | Cost minimization | cost per parcel | Total cost operational |
| | Social | Minimize waiting time | n.a. | n.a. | Minimize waiting time | Operator staff agreeableness |
| | Environment | n.a. | n.a. | $CO_2$ | $CO_2$ | $CO_2$ (IPCC based) |

## Appendix B

**Table A4.** City truck route under current conditions.

| Alternative#0 | Capacity | | Cluster #1 | Cluster #2 | Cluster #3 | Cluster #4 | Cluster #5 | Cluster #6 | Route | Load Factor |
|---|---|---|---|---|---|---|---|---|---|---|
| PMH#1 (1.5 L) | 1.1 | ton | 0.32 | | 0.32 | | 0.44 | | 0-1-3-5-0 | 97% |
| PMH#3 (1.5 L) | 1.1 | ton | | 0.30 | | 0.32 | | 0.38 | 0-2-4-6-0 | 91% |

**Table A5.** PMH routes in six clusters using the saving algorithm (three fleets).

| Alternative#1 | Capacity | | Cluster #1 | Cluster #2 | Cluster #3 | Cluster #4 | Cluster #5 | Cluster #6 | Route | Load Factor |
|---|---|---|---|---|---|---|---|---|---|---|
| PMH#1 (GB) | 0.72 | Ton | 0.32 | | | | | 0.38 | 0-6-1-0 | 97% |
| PMH#2 (GB) | 0.72 | Ton | | 0.30 | | 0.32 | | | 0-4-2-0 | 86% |
| PMH#3 (1.3 L) | 1 | Ton | | | 0.32 | | 0.44 | | 0-3-5-0 | 76% |

**Table A6.** PMH routes in six clusters using the saving algorithm (three fleets).

| Alternative#2 | Capacity | | Cluster #1 | Cluster #2 | Cluster #3 | Cluster #4 | Cluster #5 | Cluster #6 | Route | Load Factor |
|---|---|---|---|---|---|---|---|---|---|---|
| PMH#1 (1.5 L) | 1.1 | ton | 0.32 | | 0.32 | | 0.44 | | 0-1-5-3-0 | 97% |
| PMH#2 (1.5 L) | 1.1 | ton | | 0.30 | | 0.32 | | 0.38 | 0-6-2-4-0 | 91% |

**Table A7.** PMH routes in six clusters using optimation (two fleets).

| Alternative#3 | Capacity | | Cluster #1 | Cluster #2 | Cluster #3 | Cluster #4 | Cluster #5 | Cluster #6 | Route | Load Factor |
|---|---|---|---|---|---|---|---|---|---|---|
| PMH#1 (GB) | 0.72 | ton | | 0.30 | | | | 0.38 | 0-2-6-0 | 95% |
| PMH#2 (GB) | 0.72 | ton | | | 0.32 | 0.32 | | | 0-3-4-0 | 88% |
| PMH#3 (1.3 L) | 1 | ton | 0.32 | | | | 0.44 | | 0-5-1-0 | 76% |

**Table A8.** PMH routes in nine clusters using the saving algorithm (five fleets).

| Alternative#4 | Capacity | | Cluster #1 | Cluster #2 | Cluster #3 | Cluster #4 | Cluster #5 | Cluster #6 | Cluster #7 | Cluster #8 | Cluster #9 | Route | Load Factor |
|---|---|---|---|---|---|---|---|---|---|---|---|---|---|
| PMH#1 (GB) | 0.72 | ton | 0.25 | | | | | | | 0.16 | | 0-8-1-0 | 57% |
| PMH#1 (GB) | 0.72 | | | 0.18 | | | | | 0.29 | | | 0-7-2-0 | 65% |
| PMH#1 (GB) | 0.72 | | | | | | | 0.30 | | | 0.21 | 0-6-9-0 | 71% |
| PMH#2 (GB) | 0.72 | ton | | | | 0.25 | 0.23 | | | | | 0-4-5-0 | 67% |
| PMH#3 (1.3 L) | 0.72 | ton | | | 0.21 | | | | | | | 0-3-0 | 29% |

**Table A9.** PMH routes in nine clusters using the saving algorithm (three fleets).

| Alternative#5 | Capacity | | Cluster #1 | Cluster #2 | Cluster #3 | Cluster #4 | Cluster #5 | Cluster #6 | Cluster #7 | Cluster #8 | Cluster #9 | Route | Load Factor |
|---|---|---|---|---|---|---|---|---|---|---|---|---|---|
| PMH#1 (GB) | 0.72 | ton | 0.25 | 0.18 | | | | | | 0.16 | | 0-8-1-2-0 | 82% |
| PMH#2 (GB) | 0.72 | ton | | | 0.21 | | | | 0.29 | | 0.21 | 0-9-7-3-0 | 97% |
| PMH#3 (1.3 L) | 1 | ton | | | | 0.25 | 0.23 | 0.30 | | | | 0-4-5-6-0 | 78% |

**Table A10.** PMH routes in nine clusters using optimation.

| Alternative#6 | Capacity | | Cluster #1 | Cluster #2 | Cluster #3 | Cluster #4 | Cluster #5 | Cluster #6 | Cluster #7 | Cluster #8 | Cluster #9 | Route | Load Factor |
|---|---|---|---|---|---|---|---|---|---|---|---|---|---|
| PMH#1 (GB) | 0.72 | ton | | 0.18 | | | | | 0.29 | | 0.21 | 0-9-7-2-0 | 94% |
| PMH#2 (GB) | 0.72 | ton | | | | 0.25 | | 0.30 | | 0.16 | | 0-6-8-4-0 | 99% |
| PMH#3 (GB) | 0.72 | ton | 0.25 | | 0.21 | | 0.23 | | | | | 0-1-3-5-0 | 95% |

## Appendix C

**Table A11.** Ranking of 2E-SS-CVRP-PMH-SPD distribution system indicators based on delivery costs (column 2/*), SPD costs (column 3/**), and $CO_2$ (column 4/***).

| Scenario and Alternative | Economy (IDR/Item) | | Environment/ $CO_2$ (kg/Year) | Scenario and Alternative | Economy (IDR/Item) | | Environment/ $CO_2$ (kg/Year) | Scenario and Alternative | Economy (IDR/Item | | Environment/ $CO_2$ (kg/Year) *** |
|---|---|---|---|---|---|---|---|---|---|---|---|
| | Delivery * | SPD | | | Delivery | SPD ** | | | Delivery | SPD | |
| (1) | (2) | (3) | (4) | (1) | (2) | (3) | (4) | (1) | (2) | (3) | (4) |
| Scenario B Alt-2 | 3188 | 2512 | 223.18 | Scenario B Alt-1 | 3216 | 2392 | 222.69 | Scenario D | 3613 | 2779 | 19.31 |
| Scenario B Alt-3 | 3201 | 2462 | 202.51 | Scenario B Alt-3 | 3201 | 2462 | 202.51 | Scenario C | 4604 | 3542 | 19.31 |
| Scenario B Alt-1 | 3216 | 2392 | 222.69 | Scenario B Alt-6 | 3218 | 2475 | 227.88 | Scenario B Alt-3 | 3201 | 2462 | 202.51 |
| Scenario B Alt-6 | 3218 | 2475 | 227.88 | Scenario B Alt-5 | 3221 | 2478 | 229.04 | Scenario B Alt-1 | 3216 | 2392 | 222.69 |
| Scenario B Alt-5 | 3221 | 2478 | 229.04 | Scenario B Alt-2 | 3188 | 2512 | 223.18 | Scenario B Alt-2 | 3188 | 2512 | 223.18 |
| Scenario B Alt-4 | 3295 | 2602 | 235.42 | Scenario B Alt-4 | 3295 | 2602 | 235.42 | Scenario B Alt-6 | 3218 | 2475 | 227.88 |
| Scenario A | 3392 | 3084 | 413.95 | Scenario D | 3613 | 2779 | 19.31 | Scenario B Alt-5 | 3221 | 2478 | 229.04 |
| Scenario D | 3613 | 2779 | 19.31 | Scenario A | 3392 | 3084 | 413.95 | Scenario B Alt-4 | 3295 | 2602 | 235.42 |
| Scenario C | 4604 | 3542 | 19.31 | Scenario C | 4604 | 3542 | 19.31 | Scenario A | 3392 | 3084 | 413.95 |

* Ranking of 2E-SS-CVRP-PMH-SPD distribution system indicators based on delivery costs. ** Ranking of 2E-SS-CVRP-PMH-SPD distribution system indicators based on SPD costs. *** Ranking of 2E-SS-CVRP-PMH-SPD distribution system indicators based on $CO_2$.

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
