# Peer review of "E-Commerce Parcel Distribution in Urban Areas with Sustainable Performance Indicators"

_sustainability, doi:10.3390/su142316229_

Round 1

Reviewer 1 Report

This paper developed a conceptual framework for selecting the Parcel Mobile Hub (PMH) locations with sustainable performance indicators. Some suggestions are given as follows:

1.      This research uses the two terms, i.e., greenhouse gases and CO2 as the same meaning. But they are different. CO2 is the one of greenhouse gases. Please fix the problem.

2.      Literature review is very important. Please review and discuss the related papers in a section.

3.      This research includes five stages. However, the related explanation and connection of five stages are not clear. Please enhance the description.

4.      The main aim of this research focuses on selecting the Parcel Mobile Hub (PMH) locations with sustainable performance indicators. But the five stages include routing and scheduling. I think the scope of this research is unfocused.

5.      The explanation of the proposed method is not clear.

6.      Please change CO2 to CO2.     

Author Response

Thank you for the comments and suggestions. The following is an explanation of comments and suggestions from the reviewer:

  1. The purpose of this study is to build a sustainable distribution system by reducing the potential addition of CO2. We have changed the reduction of GHG to CO2.
  2. We have reviewed and discussed in the literature review section of Table A on line number 109-117.
  3. We have added an explanation and connection of five stages of research (line number 123-137).
  4. This study aims to develop a distribution system for e-commerce parcels in urban areas with sustainable indicators (line number 123-124). Routing & Scheduling is one of the stages for calculating sustainable transportation indicators (line number 132-133).
  5. The methodology used to develop an e-commerce parcel distribution system with sustainable indicators is with five stages as described in line number 123-137.
  6. We have changed CO2 to be changed to CO2.

After getting approval from the reviewers, we will send this manuscript to the editing service registered at https://www.mdpi.com/authors/english to undergo an extensive english revision.

Reviewer 2 Report

The paper adresses a very important issue of 2-echelon urban distribution strategy. It tackles with cellphone data, and census data.

One main recommendation is the native english review. The sentences are very confused and need to be rewritten in some cases. Some expressions lack the articles or the verb is not used in a correct grammatical rule.

The Figure 2 must be remade. The flows and nodes and also the caption must be better done. Figure 3 has the caption above the figure, the authors must follow the caption rules of the journal.

In section 2.3 the weight distribution of 2.5kg per packet should be better referenciated, indicating from where it came and also the range distribution of the weights took from the sample.

The first affirmative in the section 2.4 "Logistics aims..." is incorrect and imprecise, it must be rewritten.

When the autors stated that the K-means is one of the most widely used algorithms, they should point what characteristic or parameters is used to construct the clusters (geographical, # inhabitants, density of deliveries, etc).

For the description of decision variables, the language and the sentence are very confused, they should be re-written.

The authors must explain better all the optimization model equations, mainly the eq (3), and state the relationship between vol(d) and \phi(w). Also, for eq (4) why does the sum of itineraries be equal to one?

At last, one improvement should be the establishment of the links among every single part of the methodology, that is a great proposal, but it lacks a continuum line to follow.

Author Response

  1. Thank you for the comments and suggestions. The following is an explanation of comments and suggestions from the reviewer.
  2. After getting approval from the reviewers, we will send this manuscript to the editing service registered at https://www.mdpi.com/authors/english to undergo an extensive english revision.
  3. The Figure 2 has been remade with a better flows, node,s and caption (line number 145-146). The caption in Figure 3 has been transferred at the bottom of the figure (line number 173).
  4. Reference has been added in the form of research results on the top six marketplaces in Indonesia with 1240 respondents (line number 197-199)
  5. We have rewritten the first affirmative in the section 2.4 to be “For two-tier systems, tactical planning concerns the departure times, routes, and loads of urban-trucks and city-freighters, the routing of demand, and the utilization of the satellites and the distribution of work among those” (line number 201-203).
  6. The K-Means selection argument using the Silhouette coefficient is described in the line number 205-208.
  7. Description of decision variables have been improved as line number 214-216”
  8. Explanation:
    1. Equation 3, is an PMH itinerary with demand parcel (volume) must be less than PMH capacity. If the volume transported exceeds the PMH capacity, it cannot operate.
    2. The number of itineraries is equal to 1, meaning that PMH and city-freighters carry out their duties according to a predetermined schedule. For example, PMH departs from the Distribution Center at 07.00 and returns at 10.45. After that, PMH will depart back at 11.00 and return to DC at 14.45.
  9. The research methodology has been explained through five stages as described in the line number 124-136.

Reviewer 3 Report

Glad to review the manuscript (ID: sustainability-1913406). This study analyzed e-commerce parcel distribution as per different transport modes taking into consideration CO2 emissions. The study is interesting within good work effects. The feasibility and practical issues are the big concern from my perspective. The comments in detail are below to help improve further.

(1) I did not understand how to derive the 19.7% optimization. Using the minibus as Parcel Mobile Hub is an option but need to strike out whether the load capacity is available. Moreover, in the traditional mode, what is/are PMH(s)? Is the fleet size of minibusses enough to accommodate the “E-commerce transactions”? The authors should consider deeply and answer these practical questions.

(2) In addition, I did not find out the reasons why the new mode can decrease CO2 emissions. How did the authors compare them?

(3) In Table 1, some references are not derived from the date of Indonesia. Can the data of other countries be treated as the supportive gist?

(4) About the MILP model proposed, I did not think it is linear, especially checking the objective function (Eq.(1)). In addition, what are the symbols of the decision variables? How did the authors resolve the model? The details of the solution algorithm were not mentioned anymore in the context. It is a big concern. In fact, the nonlinear integer programming model, as an NP-hard problem, is not easy to generate the optimum.

(5) How did the authors conclude the point “In this study, a van/ truck with a capacity of 700kg-1,200 kg was used as the PMH. The sleek form of the Minibus with various capacities is considered the most suitable”. I did not know what are the precise capacities in detail.

(5) In terms of the calculation of greenhouse gas (CO2) emissions, are all vehicle types identical (on Pages 8-9)? It is contradictory because the minibusses are considered to decrease CO2 emissions. Where are the calculation results of the CO2 emissions?

(6) Last but not least, I carefully checked the 3 Tables A but did not find out the superiority of the current study. Thus, could please the authors introduce the merits distinctly in Tables A?

Author Response

  1. Explanation:
    1. The calculation of the 19.7% reduction in operating costs has been described in the manuscript line number 504-505.
    2. The reason for the decrease in operating costs and CO2 is due to the decrease in mileage (line number 507-508)
    3. With the volume in 2020 and 2021, the use of minibuses with a capacity of 720kg to 1.1ton is still possible. However, the type of vehicle and its capacity can be increased in line with the larger volume (line number 424 sd 425).
    4. PMH is a form of innovative distribution system, while the traditional or classic model is in the form of a static hub in the form of a building as described in line number 86-87.
  2. The decrease in CO2 is due to the new mode resulting in lower travel distances compared to the existing conditions (line 507-509).
  3. Two references from outside Indonesia, namely: there are e-buyer centers and service availability:
    1. One of the research hypotheses that will be proven is "There are e-buyer centers in Indonesia". This research has stated that the hypothesis is proven correct (line number 381-383).
    2. Practically, 24/7 service availability is run by CEP operators in Indonesia, both in the form of opening counters and customer service 24 hours a day, 7 days a week.
  4. Explanation of the Saving Algorithm and MILP model used:
    1. Equation 1 is a function of minimizing PMH and city-freighter operational costs.
    2. There are three decision variables, namely: (1) PMH work assignment, (2) city freighter work assignment, and (3) itineraries for PMH and city-freighter (t,p) (line number 215-217). Symbols and decision variables as equation 7-9 (line number 225-227).
    3. Saving Algorithm is used to perform clustering (line number 132). The solution is to use ms excel.
    4. The optimization model is used to determine the route that produces the lowest operational costs. Completion of the optimization model using ms excel.
  5. Practically, the top six operators in Indonesia use minibus vehicles for package distribution in the urban area of Bandung (Figure 7.  City-freighter used by top six CEP Operators in Indonesia in line number 660).
  6. Explanation:
    1. The amount of CO2 emissions produced by vehicles uses the formula from the IPCC 2016 (line number 311). Different types and capacities of vehicles will result in different fuel consumption which will result in different CO2 emissions. We calculate the fuel consumption for the Grandmax Blindvan (GB) vehicle with a capacity of 720 kg as much as 17.5km/ lt. Grandmax 1.3L vehicles require 15km/lt of fuel and 1.5L Grandmax require 12.5km/lt of fuel.
    2. The decrease in CO2 is not caused by the use of minibuses but the reduction in travel distance (line number 507-509). The correct statement is: “Using PMH that combines with city freighterUsing the Minibus as PMH can reduce operating costs by 19.7% and prevent additional CO2 by 3.4 tons per year with conventional motorcycles and 7.2 tons CO2 per with an electric motorcycle or scooter” (line number 21-23)
  7. An explanation of the review and discussion of previous research has been added in Table A (line number 109-117).

Notes:

After getting approval from the reviewers, we will send this manuscript to the editing service registered at https://www.mdpi.com/authors/english to undergo an extensive english revision.

Reviewer 4 Report

Thank you for the opportunity to review the paper. In general the theme of the paper is interesting. The paper tries to solve the parcel distribution problem by considering environmental factors which is an appealing effort. However, unfortunately, I found the introduction is not clear. First, it jumps from one idea to another idea. For example, I dont understand what the authors want to say in paragraph 2 of the introduction. In addition to that, I dont see a clear research aim, objectives, and questions. Also, one thing that bothers me more is the research method. It is confusing. I can't understand it. For example, I found this sentence "the new idea in this research is using spacecraft in various scenarios, namely conventional motorcycles, electric motorcycles, and scooters". Lastly, I dont see any contribution to the current knowledge. 

To sum up, I see the authors' hardwork in developing a solution. However it is poorly presented. The objective is unclear. Perhaps the authors should go back to their work, find the strongest element in their work and try to present that well. Considering the amount of re-work that the authors have to do, I regretfully have to suggest the editor to reject this paper. 

I hope the authors find my comment useful and good luck.

Author Response

Thank you for reviewing on our paper.

We apologize profusely for writing the paper in a not good English style. After getting approval from the reviewers, we will send this manuscript to the editing service registered at https://www.mdpi.com/authors/english to undergo an extensive english revision.

  1. The introduction has been revised by adding a review of previous research related to a 2-echelon distribution system with Mobile Access Hub (line number 105-117)
  2. Explanation of research questions, objectives, and aim:
    1. Research question: how to build a distribution system for two e-commerce parcels in urban areas with sustainable performance indicators (line number 119-121)
    2. Research objectives: determining the distribution system for e-commerce packages, developing a conceptual framework for selecting MAH locations, calculating sustainable transportation indicators. This goal is achieved through the five stages described in line 124-136.
    3. Research aim: to develop a distribution system for e-commerce parcels in urban areas with sustainable indicators (line number 123-124)
  3. Sorry for the typo "spacecraft" should be "city-freighter"
  4. The contribution of this paper is to fill the gaps in previous research by developing a conceptual framework for selecting PMH locations that produce indicators of sustainable transportation

Reviewer 5 Report

Check the attached document

Author Response

Thank you very much for reviewing and providing suggestions for improvement. The following is the response from the reviewer's comments:

  1. Article's originality and methodology has been added in the abstract line 13-17.
  2. The article's originality has been added in the introduction section at line 119-125.
  3. Research objectives have been added to manuscript line 127-130. While the chosen research method is described in lines 131-135, 140-141, and 149-152.
  4. We build a conceptual framework for PMH sites (model building). The city of Bandung is used as a case study to test the model. A description of the city of Bandung as a case study has been added to manuscript line 345-352.
  5. We present information on methodology in a paragraph in manuscript lines 131-152.
  6. We have added a special section (3.9 Result) to present the results generated from the research on line 532-554.
  7. The results of this study are an extension of previous research. Kami telah menambahkan implikasi teoritis di dalam manuscript pada line 563-567.

Round 2

Reviewer 2 Report

In my point of view, all the questions posted previously were fulfilled.

Author Response

Thank you for reviewing our paper. Below we attach a paper that has undergone English language editing by MDPI with a certificate (page 26).

Kind regards

Reviewer 3 Report

Dear Editor:

After checking the whole paper again, I can see the revisions were well done and responded reasonably. The report is improved much except for the language issues. Thus, please allow me to recommend a minor revision decision from my perspective. 

Best, Silin

Author Response

(The authors gave the same response as above.)
